# Sperm Parameters before and after Swim-Up of a Second Ejaculate after a Short Period of Abstinence

**DOI:** 10.3390/jcm9041029

**Published:** 2020-04-05

**Authors:** Claudio Manna, Federica Barbagallo, Raffaella Manzo, Ashraf Rahman, Davide Francomano, Aldo E. Calogero

**Affiliations:** 1Biofertility IVF and Infertility Center, 00198 Rome, Italy; claudiomanna55@gmail.com (C.M.); raffaella.manzo1987@gmail.com (R.M.); ashfarrag62@gmail.com (A.R.); 2Department of Clinical and Experimental Medicine, University of Catania, 95125 Catania, Italy; federica.barbagallo11@gmail.com; 3Altamedica ART Center, 00198 Rome, Italy; 4Unit of Internal Medicine and Endocrinology, Madonna delle Grazie Hospital, Velletri, 75100 Rome, Italy; davide.francomano@gmail.com

**Keywords:** sperm morphology, sperm motility, sperm DNA fragmentation, ART, semen analysis

## Abstract

Background: Recent studies have supported the beneficial effects of a short abstinence period on sperm parameters. The aim of this study was to assess sperm motility, morphology and DNA fragmentation before and after swim-up of a second ejaculate obtained after a short abstinence period in normozoospermic men and oligo-astheno-teratozoospermic (OAT) patients. Material and methods: Semen analyses and swim-up preparations of two consecutive semen samples (collected within 1 h) were carried out in 30 normozoospermic and 35 OAT patients enrolled in an assisted reproductive technique (ART) program. Results: Compared to the first ejaculate, the second sample showed a higher percentage of spermatozoa with normal form (*p* < 0.01) and lower percentage of spermatozoa with DNA fragmentation (*p* < 0.01) in normozoospermic men, whereas a higher percentage of spermatozoa with progressive motility (*p* < 0.001) and normal morphology (*p* < 0.0001) was found in OAT patients. Swim-up separation showed a lower DNA fragmentation rate (*p* < 0.05) in the second ejaculate in normozoospermic men, whereas the second ejaculate of OAT patents showed an increase in normally-shaped spermatozoa (*p* < 0.01) and lower percentage of spermatozoa with fragmented DNA (*p* < 0.001) compared to the first one. Conclusions: Swim-up separation of a second ejaculate collected within 1 h might be suggested for ART procedures, especially in OAT patients.

## 1. Introduction

It is well known that conventional sperm parameters (concentration, motility, and morphology) vary significantly among men and even between consecutive samples in the same man [1]. The WHO manual for sperm analysis [2] suggests that the abstinence period should range between 2 and 7 days but it does not give any hint as to the number and the frequency of previous instances of intercourse within this recommended period [2].

It is commonly believed that sperm count is inversely related to the frequency of intercourse and that, in normozoospermic men, the sperm count decreases significantly with sequential ejaculations. In fact, a consistent total sperm count decrease has been reported in the second ejaculate compared with the first one [3]. Furthermore, sperm concentration, seminal fluid volume and total sperm count decrease with frequent ejaculation, suggesting avoiding daily intercourse [4]. However, a recent study showed that with an extended 2-week period of daily ejaculation, sperm parameters remain above the WHO 2010 reference values [5]. On the contrary, a long abstinence period can lead to sperm senescence like that found in infertile patients, suggesting that it is better to obtain a second semen sample one day after the first one [6].

In cases of very poor semen fluid characteristics, a study has shown that it is better to use only a 1 h abstinence period, since the second sample was of better quality compared to the first one, suggesting that it should preferably be used for oocyte fertilization [7]. In an assisted reproductive technique (ART) setting, the fertilization rate of spermatozoa obtained from a second ejaculate was significantly higher compared with that derived from the first one, suggesting that sequential ejaculations might be useful for treatment of male infertility [8]. More recently, Bahadur and colleagues clearly showed that the “median progressive motility as benchmarked by the WHO 2010 criteria were significantly higher in the second consecutive sample (43% vs. 25%, *p* < 0.001)” in subfertile oligozoospermic patients [9]. Alipour and colleagues showed not only higher percentages of progressive motile spermatozoa but also an improvement in sperm kinematic parameters after 2 h of abstinence compared with samples obtained after 4–7 days of abstinence [10]. Specifically, spermatozoa of the second ejaculate had higher velocity (*p* < 0.001), progressiveness (*p* < 0.001) and hyperactivation (*p* < 0.001) compared with the first ejaculate [10]. Similarly, Ortiz and colleagues showed that the second ejaculate on the day of insemination leads to an improvement of the sperm sample quality in a high percentage of cases of normozoospermic men undergoing intrauterine insemination [11] A recent systematic review supports the beneficial effect of the short abstinence period on sperm parameters and recommends that the current indications on the abstinence period should be revisited [12].

The quality of ejaculated spermatozoa before and after preparation for ART procedures cannot be evaluated only by conventional sperm parameters. For this reason, different bio-functional tests have been developed and are currently used. One of the most used is the evaluation of sperm DNA fragmentation [12]. Indeed, it has been demonstrated that a high percentage of DNA-fragmented spermatozoa may be found in normozoospermic men and this might be the cause of the so-called unexplained infertility [13]. Furthermore, although oocytes are capable of repairing DNA damage [14,15], meiotic DNA damage may escape paternal repair and causes chromosomal aberrations in the zygote because of a maternal misrepair [16]. Shen and colleagues have recently found that sperm DNA fragmentation index decreased in a statistically significant manner in semen samples after a short (1–3 h) period of abstinence compared with a long (3–7 days) one. Interestingly, they also described that the rates of implantation, clinical pregnancy and live births significantly increased in the ejaculate collected after a short period of abstinence [17].

In the era of intracytoplasmic sperm injection (ICSI), it is of paramount importance to improve the selection of spermatozoa to inject into the oocyte. For this purpose, the use of the second consecutive sperm sample in oligozoospermic patients may be of value. Therefore, the aim of the present study was to evaluate conventional sperm parameters and DNA fragmentation rate before and after swim-up in two semen samples collected 1 h a part in an ART setting. 

## 2. Materials and Methods 

Sixty-five consecutive couples undergoing ICSI in a Biofertility in vitro fertilization (IVF) Centre were enrolled in the present study. Informed consent was obtained from each of the male partners. We asked all men to provide samples after having clearly explained the aim of the study: the possibility that the second ejaculate would results in better sperm parameters in the second ejaculate. All of them had an abstinence of 2–7 days, as suggested by the WHO 2010 criteria that were followed for sperm analysis (WHO, 2010). 

All semen analyses were performed by the same experienced embryologist. The evaluation of sperm motility analyses was performed on a 10 μL drop on a glass slide with a 22 × 22 mm cover slip and on a heated stage at 37 °C, with a lens that had a reticule. We examined the slide with phase-contrast optics at a magnification of 400X. We assessed 400 spermatozoa per replicate for determining the percentage of different motile categories. 

The assessment of morphology was done using Diff Quick stained slides (Medion Diagnostics AG, Bonnestrasse 9, CH-3186 Dudingen, Switzerland). Morphology was evaluated at a magnification of 400X and 1,000X. Multiple slides were prepared to obtain at least 400 spermatozoa for analyses to avoid technical difficulties in patients with severe oligozoospermia.

All semen samples were collected by masturbation within the clinic to minimize conditions that might alter sperm parameters. A second seminal fluid collection was asked for 1 h after the first, taking into account the time that each patient took to collect the first sample. 

Sperm DNA fragmentation was evaluated using the Halosperm kit (HT-HS10) using conventional bright-field microscopy [18]. This is based on the principle that sperm with fragmented DNA fails to produce the characteristic halo of dispersed DNA loops that is observed in sperm with non-fragmented DNA, following acid denaturation and the removal of nuclear proteins. A minimum of 300 spermatozoa for each sample were counted.

The ICSI procedure was performed with the best ejaculate and swim-up preparation obtained. The “swim-up” technique was performed directly from liquefied semen. To accomplish this, several aliquots of seminal fluid were taken from each sample and placed in tubes underneath an overlay of Flushing Medium (Origio Italia Srl, Rome, Italy). Round-bottom tubes or four-well dishes were used to optimize the surface area of the interface between the semen layer and the culture medium. The samples were left to incubate at 37°C in an incubator for 30–45 min. Spermatozoa with the best motility that were able to migrate were recovered.

### 2.1. Ethical Approval 

The study was conducted in the ART centre Biofertility IVF Center (Rome, Italy), on infertile couples undergoing ICSI treatment. The study was reviewed and approved by the Institutional Review Board at the Biofertilty IVF Centre, who also indicated that ethical approval was not required for this study. The project identification code is B2-2018. Data collection followed the principles outlined in the Declaration of Helsinki; all patients provided their informed consent, agreeing to supply their own anonymous information for this and future studies. 

### 2.2. Statistical Methods 

This is a within-subject analysis for assessing the influence of a very short abstinence period on sperm quality. Both samples were analysed, and the following parameters were compared: volume (ml), concentration (mil/mL), total and progressive motility (%), normal morphology (%), and DNA fragmentation (%). Nonparametric tests were used, and the statistical analysis was performed using software SPSS 17.0 (SPSS Inc., Chicago, IL USA). For paired samples, the Wilcoxon matched pairs signed-rank test was used when comparing parameters between the two ejaculates. The Kruskal–Wallis test and the chi-square or Fisher’s exact test were used to test for any associations where appropriate. Spearman’s non-parametric correlation was used to test for associations between the duration of abstinence and sperm volume. A p value lower than 0.05 was considered to be statistically significant. All results are presented as the mean (SD), median and range of proportion.

## 3. Results

This is a single-centre prospective study that enrolled 65 men. Sperm parameter readings had a low sampling error of ≤5% only. Sampling errors (%), according to the total number of spermatozoa counted, are shown in Table 2. of the WHO Manual (WHO, 2010). 

We divided the male partners of the infertile couples into two groups based on their sperm parameters. The first group had normal sperm parameters (*n* = 30; mean age: 38 ± 3.5 years), whereas the second group was made up of patients with oligo-, astheno- and/or terato-zoospermia (OAT) (*n* = 35; mean age: 37 ± 3.3 years). Only one patient with OAT was not able to provide a second semen sample after 1 h from the first one and, hence, he was excluded from the study. 

No statistically significant correlation was found between ejaculate volume and the duration of abstinence of the first semen collection.

Data of the first and second semen samples are shown in Table 1 and Table 2. Semen volume was significantly lower in the second ejaculate of both normozoospermic and OAT groups. Sperm concentration was significantly lower only in the normal group. Total and progressive motility increased significantly in the second ejaculate, but only in OAT patients. A statistically significant improvement in the sperm morphology was recorded in both normozoospermic men and OAT patients. Sperm DNA fragmentation was significantly lower in the second ejaculate of both normal and OAT groups. Table 3 and Table 4 show sperm parameters and the percentage of spermatozoa with DNA fragmentation separated by swim-up. Sperm concentration as significantly lower in the supernatant of the second ejaculate in both groups. Normal sperm morphology of the second sample was significantly higher in normozoospermic men and in OAT patients. A lower percentage of sperm fragmentation was seen in normal men and it was greatly decreased in OAT patients. The latter percentage decrement in the second sample was much higher in OAT patients compared with that found in the normozoospermic men. Only six of the second seminal samples out of 65 (9.2%) were not used for the ICSI procedure. In fact, in these cases, the first ejaculate was better than the second one. The embryologist of our centre was free to decide which sample was best to use for ART.

## 4. Discussion

This study showed that the second consecutive ejaculate resulted in better conventional sperm parameters (motility and morphology) and in a lower percentage of spermatozoa with fragmented DNA in normozoospermic male partners of infertile couples and even more in patients with OAT. Precisely, in men with normozoospermia, this trend was significant only for spermatozoa with a normal morphology and DNA fragmentation before and after swim-up preparation. The second semen sample of patients with OAT showed better total and progressive sperm motility, as well as normally shaped spermatozoa. Interestingly, after swim-up separation, the percentage of DNA-fragmented spermatozoa decreased significantly in patients with OAT. Similarly, a decreased sperm fragmentation index in the swim-up fraction was described by Younglai and colleagues, but this study evaluated only normozoospermic semen samples [19]. On the other hand, a lower percentage of DNA-fragmented spermatozoa in semen samples collected after short abstinence periods has already been reported [20] as well as the presence of a correlation between sperm DNA integrity, progressive motility and fertilization [21]. We found a greater decrease in the percentage of DNA-fragmented spermatozoa in patients with OAT after swim-up compared with the less pronounced increase in the percentage of motile and normally shaped spermatozoa in the same samples. These findings suggest that swim-up separation is able to select spermatozoa with better DNA integrity, motility and morphology from the second consecutive ejaculate in OAT patients.

The mechanism(s) by which a better sperm “quality” was found in a second ejaculate collected after a short interval is unclear. A short period of abstinence could result in a “younger” population of spermatozoa that have a lower period of exposure to the toxic effects of reactive oxygen species (ROS) [22]. The improvement may also depend on the increase in the total antioxidant capacity (TAC) of semen and lower sperm membrane lipid peroxidation (LPO), as recently reported [9,23,24]. This hypothesis is further supported by the decrease in the percentage of spermatozoa with fragmented DNA in semen samples collected after shorter abstinence periods [17,25]. Indeed, Hussein and colleagues found a significant improvement in DNA integrity in the second semen sample collected after 1–3 h to the first one from 20 infertile patients with idiopathic OAT [25]. Shen and colleagues described that sperm DNA fragmentation index significantly decreased (*p* < 0.05) in semen samples collected after 1–3 h of abstinence compared with that of spermatozoa collected after a long period (3–7 days) [17]. 

The increment of the sperm motility observed in the second ejaculate could be explained by a different epididymal transit time. Sperm maturation during the epididymal transit involves sperm surface modifications and changes of flagellar beating by which spermatozoa acquire their forward motility [26]. In addition, epigenetic modifications occur during the epididymal transit. [27]. The second ejaculate transits through the cauda more quickly compared to the first one, resulting an increased motility potential. This may be explained by the fact that the maximum sperm motility is acquired in the corpus proximal to the cauda. After emptying the cauda with the first ejaculation, spermatozoa coming from the corpus of the epididymis are ejaculated next and this may explain the motility improvement [11]. Shen and colleagues used a proteomic approach to confirm the potential molecular difference of spermatozoa in ejaculates after 3–7 days and 1–3 h of abstinence. They found that proteins highly involved in sperm motility and capacitation were differentially expressed and that the acrosome reaction capability of spermatozoa was markedly higher after 1–3 h [17]. Recently, cyclooxygenase 1 (COX-1) was indicated as a biomarker that may identify spermatozoa of a better quality [28]. Liu and colleagues analysed two consecutive semen samples collected at 1 h intervals from 48 men. They found that COX-1 levels correlate positive with semen volume, sperm concentration and progressive motility, whereas a negative correlation was found between the rate of sperm DNA fragmentation and COX-1. Although the quality of the first sample was better compared with the second one and this is in contrast with our results, the cut-off of COX-1 (<17.5 ng/mL), as an indicator to evaluate sperm quality, does not seem to be limited by sampling time and the results were similar in both first and second ejaculates [28].

Bahadur and colleagues reported the increase in sperm progressive motility, morphology and concentration in 73 sub-fertile patients with short abstinence periods (up to 40 min). Regarding the possible reasons for these improvements, the authors discussed also about the biochemical changes between the two consecutive ejaculates [9]. The decrease in Na^+^ and Ca^2+^ concentrations and the decrease in the cytoplasmic pH in spermatozoa from the caput to the cauda of the epididymis [29] could be involved in the increase in sperm motility in the second ejaculate.

Our data stimulate some considerations on both natural and assisted reproduction. The first consideration is relative to couples of reproductive age. The findings of this study suggest that more frequent intercourse in a very short period of time could enhance conception. This consideration seems in contrast with the conventional idea of choosing the most fertile day of the menstrual cycle for a single period of intercourse and suggesting, at the same time, to refrain from sexual intercourse in the days prior to improve total sperm count. These indications might be even more improper for sub-fertile couples due to the male factor. In fact, our data are in line with the most recent literature suggesting the opposite: the second ejaculation shows overall better sperm quality in terms of motility, normal morphology and DNA integrity [7,9,10,11,17,25,30,31,32]. This evidence becomes extremely relevant in OAT patients. These observations add further explanation to the low relationship between conventional sperm parameters and fertility possibly related to the heterogeneity of the human semen enhanced by the variable length of the abstinence period, which it is suggested should range from two to seven days [2]. The WHO manual is, in fact, based on data of male partners of fertile couples without data on sexual intercourse frequency. It is possible that the suboptimal sperm parameters of the first intercourse may improve in a subsequent one some hours later. According to Ortiz and colleagues, many of the first sperm ejaculates, collected in an intrauterine insemination (IUI) setting considered suboptimal became normal in a second subsequent ejaculate [11].

It is known that more suitable spermatozoa for fertilization are those physiologically selected from the female reproductive tract during intercourse in a sort of natural “swim-up” procedure [33]. This strongly reinforces our findings, showing that the second sperm swim-up separation resulted particularly performing. These findings suggest possible consequences in ART procedures. In IUI or ART, for example, it could be suggested to separate spermatozoa by the swim-up of a second ejaculate, particularly if the first one results in suboptimal parameters. In fact, Bahadur and colleagues [34] showed that recurrent ejaculates successfully improved IUI pregnancy rates and that reduced male abstinence to 12–24 h was associated to improved pregnancy rate with ICSI [34]. In the latter case, after swim-up with the second consecutive semen sample, the oocyte can be injected with spermatozoa of better quality, considering the increased probability of choosing a gamete with intact DNA among the many deemed morphologically “normal” by the embryologist. With IVF-ICSI procedures, the improvement of sperm quality enhances fertilization rate and embryo quality, as already shown [8]. This possible trend might be particularly important in infertile couples with female advanced age. In fact, the injection of less DNA-damaged spermatozoa may be useful considering the lower DNA repairing capability of oocytes of older women [18]. In fact, a correlation between DNA sperm fragmentation and aneuploidy was found in normozoospermic men [35]. A comprehensive strategy to adopt in the male partners of infertile couples, especially if they have OAT, could be represented by a proper diagnostic and therapeutic work-up associated to short abstinence period to maximize the quality of spermatozoa to inject for oocyte fertilization.

### 4.1. Limitations of the Study

The results of this study have been obtained in a single centre and with a relatively low number of men. To corroborate these findings, larger trials in IUI and IVF-ICSI cycles should be set up with prospective and randomized protocols. 

### 4.2. Future Applications and Suggestions 

Studies supporting the beneficial effects of a short abstinence period on sperm parameters have grown in recent years. Nevertheless, the possibility of using the second semen sample obtained shortly after the first one in ART programs is not yet considered in clinical practice. Based on the results of the present study, it is reasonable to consider the use of the second sample, especially in patients with OAT. If a simple ovarian stimulation is planned, two periods of intercourse within a short interval could be suggested. Furthermore, a swim-up procedure after short abstinence might be used routinely in IUI and IVF-ICSI programs. Finally, these findings can stimulate the discussion on the present abstinence length suggested in ART settings and “in-vivo” for natural reproduction. 

## Figures and Tables

**Table 1 jcm-09-01029-t001:** Seminal fluid volume, conventional sperm parameters and DNA fragmentation of the two ejaculates in men with normal sperm parameters (*n* = 30).

	First Ejaculate		Second Ejaculate		Pair-Wise Differences Mean (SE); Median
	Mean ± SD	Median (25–75%)	Mean ± SD	Median (25–75%)	
Volume (mL)	3.1 ± 1.7	3.0 (1.5–3.9)	1.7 ± 0.8	1.5 (1.0–2.0)	−1.4 ± 0.2; 1.5 ***
Concentration (mil/mL)	99.5 ± 54.1	90 (60–120)	71.5 ± 61.5	30.0 (5.0–60)	−28.0 ± 0.1; −60.0 **
Total Motility (%)	70 ± 6.78	70 (65–75)	71.8 ± 9.4	70.0 (67.5–80)	1.8 ± 0.3; 0.0
Progressive Motility (%)	46.7 ± 9.1	40 (35–50)	47.1 ± 12.8	45.0 (30–50)	0.4 ± 1.5; 5.0
Normal Morphology (%)	20.4 ± 7.1	18 (11–25)	22.4 ± 7.8	20.0 (11.5–30)	1.9 ± 1.2; 2.0 *
DNA Fragmentation (%)	14.8 ± 8.6	14 (8.75–20)	13.6 ± 9.6	13.0 (6.0–19.3)	−1.8 ± 1.2; −1.0 *

*** *p* <0.0001; ** *p* < 0.001; * *p* < 0.01.

**Table 2 jcm-09-01029-t002:** Seminal fluid volume, conventional sperm parameters and DNA fragmentation of the two ejaculates in patients with oligo-astheno- and/or terato-zoospermia (*n* = 35).

	First Ejaculate		Second Ejaculate		Pair-Wise Differences Mean (SE); Median
	Mean ± SD	Median (25–75%)	Mean ± SD	Median (25–75%)	
Volume (mL)	2.8 ± 1.1	2.5 (1.5–2.5)	1.8 ± 0.8	1.5 (0.5–1.5)	−1.0 ± 0.1; −1.0 ***
Concentration (mil/mL)	44.6 ± 40.1	12.0 (2.5–82)	42.7 ± 35.5	11.1 (2.0–62.5)	−1.9 ± 0.5; −0.9
Total Motility (%)	34.8 ± 16.4	33.5 (20–46)	49.4 ± 15.2	50.0 (40–60)	14.6 ± 0.2; 16.5 ***
Progressive Motility (%)	16.9 ± 13.6	15.0 (7.3–25)	27.5 ± 10.6	27.5 (20–35)	10.7 ± 0.4; 12.5 ***
Normal Morphology (%)	13.0 ± 6.8	10.0 (8.0–18)	15.6 ± 8.5	13.5 (9.5–20)	2.6 ± 0.3; 3.5 **
DNA Fragmentation (%)	28.3 ± 16.8	22.0 (15–35)	25.2 ± 16.0	20.0 (15–34)	−3.2 ± 0.2; −2.0 *

*** *p* < 0.0001; ** *p* < 0.001; * *p* < 6770.01.

**Table 3 jcm-09-01029-t003:** Conventional sperm parameters and DNA fragmentation of two consecutive ejaculates of normozoospermic men separated by swim-up (*n* = 30).

	First Ejaculate		Second Ejaculate		Pair-Wise Differences Mean (SE); Median
	Mean ± SD	Median (25–75%)	Mean ± SD	Median (25–75%)	
Concentration (mil/mL)	37.6 ± 28.7	25.0 (10–55)	23.5 ± 25.7	10.0 (5.5–32.5)	−14.1 ± 0.50; −15.0 **
Progressive Motility (%)	80.8 ± 10.7	80.0 (65–90)	80.1 ± 15.4	80.0 (70.0–95)	−0.68 ± 0.09; 0.0
Normal Morphology (%)	25.0 ± 10.0	21.0 (12.0–30)	27.2 ± 18.2	22.0 (12–30)	2.3 ± 1,.2; 1.0 *
DNA Fragmentation (%)	12.7 ± 8.9	9.0 (6.0–15)	11.2 ± 9.1 *	10.0 (4.3–15)	−1.5 ± 0.5; 1.0 *

** *p* < 0.001; * *p* < 0.05.

**Table 4 jcm-09-01029-t004:** Conventional sperm parameters and DNA fragmentation of two consecutive ejaculates of patients with oligo, astheno- and/or terato-zoospermia separated by swim-up (*n* = 35).

	First Ejaculate		Second Ejaculate		Pair-Wise Differences Mean (SE); Median
	Mean ± SD	Median (25–75%)	Mean ± SD	Median (25–75%)	
Concentration (mil/mL)	18.1 ± 18.2	8.5 (4.3–30)	17.4 ± 18.8	12.0 (5.3–21.3)	−0.7± 0.5; 3.5 **
Progressive Motility (%)	71.7 ± 23.3	77.5 (60.0–93.8)	76.6 ± 21.6	80.0 (65.0–95.5)	5 ± 0.4; 2.5 *
Normal Morphology (%)	24.1 ± 27.4	10.0 (8–30)	31.0 ± 32.9	15.0 (10–30)	6.9 ± 0.7; 5.0 *
DNA Fragmentation (%)	21.6 ± 10.6	20.0 (13.5–30)	16.7 ± 8.4	17.0 (10–20)	−4.9 ± 0.8; −3.0 **

** *p* < 0.001; * *p* < 0.01.

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
