# Peer review of "Sperm Parameters before and after Swim-Up of a Second Ejaculate after a Short Period of Abstinence"

_jcm, 2020, doi:10.3390/jcm9041029_

Round 1
Reviewer 1 Report
This present article suggests that sperm motility parameters, morphology and DNA fragmentation index are improved in the 2nd ejaculate following the previous 1st ejaculate, 1 hour before. This is especially true for OAT patients. Although the novelty of the article is relatively low, it manifests the results of several previous articles that suggest a similar strategy to improve ART outcome for OAT patients.
The introduction and discussion are thoroughly done and have a good structure. The results section seems a bit short. Do you know anything about the motility duration of the sperm in the 2 ejaculates? Any expectation or data for the outcome of ART from using the 2nd ejaculate from OAT patients?
Materials & Methods: It would be helpful to know more details on the motility analysis: what microscope, camera, objective, frame rate, frame number, software was used to obtain the results? More details on the morphological analysis or references would be important as well.
Was the study performed as a blinded experiment? Especially, since the results were obtained by a single experienced embryologist, how did you avoid a biased image analysis?
A few small remarks:
page 3, l.103: for each or "per"
p.7, l.10: Please explain the correlation between DNA fragmentation and motilty /morphology.
p.7, l.20: please rephrase this sentence: "This suggests that the second ejaculate encompasses spermatozoa of a better quality than that conventional parameters are not able to show."
p.7, l.36: is it really the length of transit that is different or is it the time period?
p.8, l.68:in "the" second ejaculate
Overall, a nice study with outcome that supports previous studies and suggests to use the 2nd ejaculate especially in OAT patients in routine clinical practices in ART.
Author Response
Answers to the Reviewer #1 comments
Manuscript n. jcm-760181– Revised - Sperm Parameters Before And After Swim-Up Of A Second Ejaculate After A Short Period Of Abstinence
This present article suggests that sperm motility parameters, morphology and DNA fragmentation index are improved in the 2nd ejaculate following the previous 1st ejaculate, 1 hour before. This is especially true for OAT patients. Although the novelty of the article is relatively low, it manifests the results of several previous articles that suggest a similar strategy to improve ART outcome for OAT patients.
1.The introduction and discussion are thoroughly done and have a good structure. The results section seems a bit short.
1.a. Do you know anything about the motility duration of the sperm in the 2 ejaculates?
Answer to comment 1a: Although sperm motility can be time dependent and it may decline after some hours from the collection, many studies have shown that repeated measures of sperm motility is often unnecessary. Therefore, we do not have any data on sperm motility duration of the second ejaculate. According to the WHO 2010 criteria, we evaluated sperm motility preferably at 30 minutes, but in any case within 1 hour, from semen collection for both ejaculates and we found that sperm motility of the second ejaculate was significantly higher than that of the first one. Data are shown in Tables 1 and 2.
1.b. Any expectation or data for the outcome of ART from using the 2nd ejaculate from OAT patients?
Answer to comment 1b: The embryologist of our centre was left free to decide which sample was better to use for ART and in 90.8% of cases, he decided to use the second ejaculate because its parameters were better than those of first one. Using the second ejaculation, we had a live birth rate of ~25%. This was slightly higher compared to the one obtained using spermatozoa from ejaculates collected after the abstinence period suggested by the WHO 2010 manual. However, we cannot tell whether this increase was statistically different because these latter are retrospective results of our ART Centre. For this reason, based on these results, we are planning an ad hoc study which will also evaluate ART outcome (fertilization rate, embryo quality, pregnancy rate, and live birth rate) using spermatozoa from the first and second ejaculate by splitting the oocytes when they are retrieved in an adequate number. This study is under Ethical Committee evaluation.
- Materials & Methods: It would be helpful to know more details on the motility analysis: what microscope, camera, objective, frame rate, frame number, software was used to obtain the results?
Answer to comment 2: We did not use any software to evaluate sperm motility. The assessment of this parameter was performed by an expert seminologist (always the same one) according to the WHO 2010 criteria. In brief, motility analyses were performed on a 10 µl drop on a glass slide with a 22 × 22 mm cover slip and on a heated stage at 37°C, and the lens had a reticule. We examined the slide with a phase-contrast optics at ×400 magnification and we assessed 200 spermatozoa per replicate for the percentage of different motile categories.
This has been added to Materials & Methods (please see lines 88-92).
More details on the morphological analysis or references would be important as well.
Answer to comment 2, part on morphology: The assessment of morphology was performed using the Diff Quick stained slides (Medion Diagnostics AG, Bonnestrasse 9, CH-3186 Dudingen, Switzerland). Morphology was evaluated at a magnification of 400X and 1,000X. Multiple slides were prepared to obtain at least 400 spermatozoa for analyses to avoid technical difficulties in patients with severe oligozoospermia. The evaluation was done according the WHO 2010 manual. Briefly, it comprised the following steps:
- Preparing a smear of semen on a slide
- Air-drying, fixing and staining the slide
- Mounting the slide with a coverslip if the slide is to be kept for a long time
- Examining the slide with brightfield optics at ×1000 magnification with oil immersion
- Assessing approximately 200 spermatozoa per replicate for the percentage of normal forms or of normal and abnormal forms
- Comparing replicate values to see if they are acceptably close: if so, proceeding with calculations; if not, re-reading the slides.
This methodology has been added to Materials & Methods (please see lines 94-97)
- Was the study performed as a blinded experiment? Especially, since the results were obtained by a single experienced embryologist, how did you avoid a biased image analysis?
Answer to comment 3: All semen analyses were performed by the same experienced embryologist. The protocol could not be designed in a blinded manner because the embryologist in charge to decide which sample was better to use for ART after having evaluated both ejaculates. Therefore, he was left free to decide where spermatozoa for ICSI had to be taken from.
- A few small remarks:
- page 3, l.103: for each or "per". Changed as suggested (please see page 3, line 105).
- 7, l.10: Please explain the correlation between DNA fragmentation and motility/morphology.
- We thank the Referee for having pointed this out. We were not clear in writing this sentence. This is what we think on this topic.
- Previous evidence has shown that sperm DNA integrity is highly compromised in patients with extremely low sperm count, poor motility, and/or high percentage of spermatozoa with an abnormal morphology (Sun et al., 1997). Nevertheless, other studies did not find any correlation between sperm DNA fragmentation and conventional sperm parameters, suggesting that the evaluation of sperm DNA fragmentation provides clinically relevant information for natural or ART, independently of those derived from conventional semen parameters (Condorelli et al., 2020). The lower percentage of spermatozoa with fragmented DNA that we found in the second ejaculate, associated with an increased sperm motility and the higher percentage of spermatozoa with normal morphology, may relate to the very short abstinence period. In fact, as reported in previous studies, a second ejaculation may be associated with spermatozoa of “better quality”, in terms of both conventional parameters and DNA fragmentation. The exactly mechanism(s) is not clear, but in Discussion we have examined some hypotheses. Among these, the hypothesis of a “younger” population of spermatozoa, with a lower period of exposure to the toxic effects of reactive oxygen species (ROS), is perfectly in line with the decrease of the percentage of spermatozoa with fragmented DNA in semen samples collected after shorter abstinence.
- Therefore, this sentence was deleted in the revised version of the manuscript.
- 7, l.20: please rephrase this sentence: "This suggests that the second ejaculate encompasses spermatozoa of a better quality than that conventional parameters are not able to show."
The sentence was change in: “This suggests that the evaluation of conventional parameters alone is not sufficient to show the better sperm quality found in the second ejaculate” (please see page 7, lines 20-21).
- 7, l.36: is it really the length of transit that is different or is it the time period?
The mistake was corrected (please see page 7, lines 31-32).
- 8, l.68:in "the" second ejaculate.
Done as suggested (please see page 8, line 56).
- Overall, a nice study with outcome that supports previous studies and suggests to use the 2ndejaculate especially in OAT patients in routine clinical practices in ART.
Thank you for this comment.
Reviewer 2 Report
The manuscript entitled “Sperm Parameters Before And After Swim-Up Of A Second Ejaculate After A Short Period Of Abstinence” by Manna et al. shows that the sperm from the second consecutive ejaculates is better quality than that from the first ejaculate especially on the oligo-/astheno-/terato-zoospermic patients.
It is interesting idea and give some new aspects for ART technology.However, the manuscript is not the first report to the idea, as explained by authors in Discussion.
I feel it is not sufficient to examine the halosperm for evaluate DNA fragmentation. Furhtermore, the author did not examine whether the sperm from the second consecutive ejaculate was really good of ART including ICSI.
However, the manuscript still has some important new knowledge that quality of sperm of the OAT patients was increased by collection after short abstinence and the swim-up treatment.
I feel it is better to revise some points will be revised before accept. Re-review is not required.
Specific points:
- Some citation marks are wrong feature (for example [20], [21] in page 7).
- 3rd and 4th paragraph on page 7: Discussion why sperm quality of the second ejaculate was better than the first ejaculate. These are interesting discussion but I feel these are over discussion, because the author did not show any results about oxidation status of the sperm and location of the sperm before ejaculation. It is better to describe concisely.
Author Response
Answers to the Reviewer #2 comments
Manuscript n. jcm-760181 – Revised - Sperm Parameters Before And After Swim-Up Of A Second Ejaculate After A Short Period Of Abstinence
The manuscript entitled “Sperm Parameters Before And After Swim-Up Of A Second Ejaculate After A Short Period Of Abstinence” by Manna et al. shows that the sperm from the second consecutive ejaculates is better quality than that from the first ejaculate especially on the oligo-/astheno-/terato-zoospermic patients.
It is interesting idea and give some new aspects for ART technology. However, the manuscript is not the first report to the idea, as explained by authors in Discussion.
I feel it is not sufficient to examine the halosperm for evaluate DNA fragmentation. Furthermore, the author did not examine whether the sperm from the second consecutive ejaculate was really good of ART including ICSI.
However, the manuscript still has some important new knowledge that quality of sperm of the OAT patients was increased by collection after short abstinence and the swim-up treatment.
I feel it is better to revise some points will be revised before accept. Re-review is not required.
Specific points:
1. Some citation marks are wrong feature (for example [20], [21] in page 7).
We have corrected the mistakes (please see page 7, lines 13-14).
2. 3rd and 4th paragraph on page 7: Discussion why sperm quality of the second ejaculate was better than the first ejaculate. These are interesting discussion but I feel these are over discussion, because the author did not show any results about oxidation status of the sperm and location of the sperm before ejaculation. It is better to describe concisely.
We have described some concepts in a more concise way (please see page 7, lines 31-50).
Reviewer 3 Report
In this manuscript Manna et al., show the analysis for assessing the influence of a very short abstinence period on sperm quality. They evaluate conventional sperm parameters and DNA fragmentation rate before and after swim-up in two semen samples collected 1h a part in an ART setting. They show that the second consecutive ejaculate resulted in better conventional sperm parameters (motility and morphology) and in a lower percentage of spermatozoa with fragmented DNA in normozoospermic male partners of infertile couples and even more in OAT patients.
This manuscript provides an interesting result on how get better seminal samples in OAT patients. It shows that the second ejaculated in OAT samples after swmin-up, the seminal characteristics (motility, morphology and DNA fragmentation) get almost arrive to levels of normozoospermic samples.
The sample is small, but this results support to others studies show the beneficial effect of the short abstinence period on sperm parameters and recommends that the current indications on the abstinence period should be revisited. Therefore this results are very relevant to get better the sperm sample in male with problems of fertility.
The results are well discussed, however the part of Material and Methods are shallowly explained, with little detail. The swing-up procedure should be described better, such as time, temperature, incubator, culture medium (which),etc.
Minor remarks
-There are abbreviations used several times and they aren’t described. For instance IUI.
- Line138: data of FIRST and second semen samples are shown in Tables 1 and 2. In all tables comparing the two samples.
- Explain acronyms such as: IUI
-Discussion: in lines 16,17, 87 and 101: Small mistakes in reference numbers ( 20 ,21, 33 y 34 are written like super-index)
- There are references in different formats. Such as from the reference number 22 to the 27 one. There are references with the doi underlined and others without underlining.
-Improve table formatting.
Author Response
Answers to the Reviewer #3 comments
Manuscript n. jcm-760181 – Revised - Sperm Parameters Before And After Swim-Up Of A Second Ejaculate After A Short Period Of Abstinence
In this manuscript Manna et al., show the analysis for assessing the influence of a very short abstinence period on sperm quality. They evaluate conventional sperm parameters and DNA fragmentation rate before and after swim-up in two semen samples collected 1h a part in an ART setting. They show that the second consecutive ejaculate resulted in better conventional sperm parameters (motility and morphology) and in a lower percentage of spermatozoa with fragmented DNA in normozoospermic male partners of infertile couples and even more in OAT patients. This manuscript provides an interesting result on how get better seminal samples in OAT patients. It shows that the second ejaculated in OAT samples after swim-up, the seminal characteristics (motility, morphology and DNA fragmentation) get almost arrive to levels of normozoospermic samples. The sample is small, but this results support to others studies show the beneficial effect of the short abstinence period on sperm parameters and recommends that the current indications on the abstinence period should be revisited. Therefore, this results are very relevant to get better the sperm sample in male with problems of fertility.
Comment 1: The results are well discussed, however the part of Material and Methods are shallowly explained, with little detail. The swing-up procedure should be described better, such as time, temperature, incubator, culture medium (which), etc.
Answer to comment 1: Thank you for this comment. In the revised version of the manuscript the section on Materials & Methods was enriched with the description of the method used to evaluate sperm motility and morphology as well as the swim-up technique (please see page 3, lines 88-92; page 3, lines 94-97; and page 3, lines 107-112).
Minor remarks
- There are abbreviations used several times and they aren’t described. For instance, IUI.
Done as suggested (please see page 8, line 73).
- Line 138: data of FIRST and second semen samples are shown in Tables 1 and 2. In all tables comparing the two samples.
Corrected (please see page 4, line 142).
- Explain acronyms such as: IUI
Done (please see page 8, line 73).
- Discussion: in lines 16,17, 87 and 101: Small mistakes in reference numbers (20 ,21, 33, 34 are written like super-index)
Typo mistakes were corrected (please see line page 7, lines 13 and 14 and page 8, lines 76, 82 and 90).
- There are references in different formats. Such as from the reference number 22 to the 27 one. There are references with the doi underlined and others without underlining.
We formatted all the references in the same way, according to the instruction of J Clin Med.
- Improve table formatting.
As suggested, we improved table formatting.